# LARGE MARGIN NEURAL LANGUAGE MODELS

## ABSTRACT

Neural language models (NLMs) are generative, and they model the distribution of grammatical sentences. Trained on huge corpus, NLMs are pushing the limit of modeling accuracy. Besides, they have also been applied to supervised learning tasks that decode text, e.g., automatic speech recognition (ASR). By re-scoring the $n$-best list, NLM can select grammatically more correct candidate among the list, and significantly reduce word/char error rate. However, the generative nature of NLM may not guarantee a discrimination between "good" and "bad" (in a task-specific sense) sentences, resulting in suboptimal performance. This work proposes an approach to adapt a generative NLM to a discriminative one. Different from the commonly used maximum likelihood objective, the proposed method aims at enlarging the *margin* between the "good" and "bad" sentences. It is trained *end-to-end* and can be widely applied to tasks that involve the re-scoring of the decoded text. Significant gains are observed in both ASR and statistical machine translation (SMT) tasks.

## 1 INTRODUCTION

Language models (LMs) estimate the likelihood of a symbol sequence $\{s_i\}_{i=0}^n$, based on the joint probability,

$$p(s_0, \ldots, s_n) = p(s_0) \prod_{i=1}^{n} p(s_i|s_{i-1}, s_{i-2}, \ldots, s_0). \tag{1}$$

Perplexity (PPL) is a commonly adopted metric to measure the quality of an LM. It is exponentiated per-symbol negative log-likelihood,

$$\text{PPL} \stackrel{\text{def}}{=\!=} \exp\left\{-\mathbb{E}\left[\log p(s_i|s_{i-1}, s_{i-2}, \ldots, s_0)\right]\right\},$$

where the expectation $\mathbb{E}$ is taken with respect to all the symbols. A good language model has a small PPL, being able to assign higher likelihoods to sentences that are more likely to appear.

*N-gram* models (Chen & Goodman, 1996) assume that each symbol depends on the previous $N - 1$ symbols. This restrictive assumption is also seen in LMs that are based on feed forward network (Bengio et al., 2003). To model longer-term dependencies, recurrent neural networks (e.g., Mikolov et al., 2010) are adopted. Recurrent neural language models (NLMs) often achieve smaller PPLs than *N-gram* models (Sundermeyer et al., 2012; Zaremba et al., 2014; Jozefowicz et al., 2016). More recently, alternative neural architectures (e.g., Dauphin et al., 2016) are proposed, aiming at improving training efficiency.

LMs are widely applied in automatic speech recognition (ASR) (Yu & Deng, 2014) and statistical machine translation (SMT) (Koehn, 2010). Following (Koehn, 2010), one may interpret the language model as a prior knowledge on the text to be inferred, which provides complementary information in addition to the ASR or SMT system itself. In practice, there are several ways to incorporate the language model. The simplest way may be re-scoring $n$-best list returned by the ASR or SMT system (Mikolov et al., 2010; Sundermeyer et al., 2012). A slightly more complicated way is to jointly consider the ASR/SMT and language model in a beam search decoder (Amodei et al., 2016). Specifically, at each time step, the decoder appends every symbol in the vocabulary to each sequence in the current candidate set. For every hypothesis, a score is calculated as a linear combination of the log-likelihoods given by both the ASR/SMT and language model. Then, only the top $n$ hypotheses with the highest scores are retained, as an updated candidate set. More recent

works, Gulcehre et al. (2015) and Sriram et al. (2017), propose to predict the next symbol based on a fusion of the hidden states in the ASR/SMT and language model. A gating mechanism is jointly trained to determine how much the language model should contribute.

The afore-discussed language models are generative in the sense that they merely model the joint distribution of a symbol sequence (Eq. (1)). While the research community is mostly focused on pushing the limit of modeling accuracy (lower PPL) (e.g., Jozefowicz et al., 2016), very limited attention has been paid to the discrimination ability of language models when they are applied to supervised learning tasks, such as ASR and SMT. Discriminative language modeling aims at enhancing the performance in supervised learning tasks. In specific, existing works (Roark et al., 2004; 2007; Peng et al., 2013) often target at improving ASR accuracy. The key motivation underlying them is that the model should be able to discriminate between "good" and "bad" sentences in a task-specific sense, instead of just modeling grammatical ones. The common methodology is to build a binary classifier upon hand-crafted features extracted from the sentences. However, it is not obvious how these methods can utilize large unsupervised corpus, which is often easily available, and the hand-crafted features are also ad hoc and may result in suboptimal performance.

In this work, we study how to improve the discrimination ability of a neural language model. The proposed method enlarges the difference between the log-likelihoods of "good" and "bad" sentences. In contrast to the existing works (Roark et al., 2004; 2007; Peng et al., 2013), our method does not rely on hand-crafted features. It is trained in end-to-end manner and able to take advantage of large external text corpus. We apply the proposed large margin language model to ASR and SMT tasks. It reduces word error rate (WER) and increases bilingual evaluation understudy (BLEU) scores significantly, showing notable advantage over several alternative methods that are well adopted.

## 2   RELATED WORK

Roark et al. (2004; 2007) and Peng et al. (2013) proposed to train discriminative language models based on hand crafted features. They essentially build linear classifiers that give high scores on "good" sentences but low scores on "bad" ones. These methods all rely on ad hoc choice of features, e.g., counts of $n$-grams where $n$ varies in a small range (e.g., $1 \sim 3$). Moreover, it is also not clear how these methods would take advantage of an existing language model (trained on large unsupervised corpus). Tachioka & Watanabe (2015) tries to overcome the above issues by adapting an NLM on the transcriptions of a speech dataset. Although the setup is more similar to ours, their objective is not well-behaved and difficult to optimize when there are multiple beam candidates. An in-depth discussion will be given in Section 3.1.

Kuo et al. (2002) designed another approach to train a discriminative language model, which is based on bi-grams. Similar to our method, the objective there aims at increasing the difference between the scores of the best candidate and ground-truth. However, since the language model is not end-to-end, there are several issues complicating the training, e.g., handling back-off weight.

Our proposed method is based on comparisons between pairs of sentences. Its implementation resembles siamese network architecture (Chopra et al., 2005; Taigman et al., 2014), first proposed for face verification tasks. Recently, siamese network has also been applied to learning similarities on sequences (Mueller & Thyagarajan, 2016; Neculoiu et al., 2016). In spite of solving different problems, the common methodology is to extract a pair of hidden representations for a pair of input samples (through a shared network). It then manipulates the distance between the hidden representations based on whether the two samples are considered similar or not. Our work also draws some inspirations from information retrieval (IR) (Liu, 2009). As a representative IR method, ranking SVM (Herbrich et al., 2000) assumes a linear scoring function, and imposes a hinge loss on the difference between the scores of sample pairs.

## 3   PROBLEM FORMULATION

We explicitly define LM *score* as the log-likelihood of a sentence estimated by an NLM. Existing works on NLM often train to maximize the score on a corpus that are assumed to be grammatical. However, they do not utilize any negative examples, where the "negative" means incompetence for a specific task. For example, in ASR (Amodei et al., 2016), negative samples may have spelling

or grammar errors. In a conversational model (Vinyals & Le, 2015), negative samples are non-informative replies like "I don't know". An NLM trained in the maximum-likelihood fashion is not aware of the specific task. It therefore cannot guarantee a larger score for a positive sample than a negative one. This fact may handicap applications that need LMs to distinguish between positive and negative candidates. Examples include automatic speech recognition (ASR) and statistical machine translation (SMT). We aim at enhancing the LM's discrimination in these applications.

Interestingly, negative samples are easy to obtain in the aforementioned applications. A beam search decoder can often yield abundant negative sentences (suboptimal beam candidates) that differ from ground-truth text (considered as positive). We motivate our method with an ASR example. A CTC-based (Graves et al., 2006) ASR model is trained on Wall Street Journal (WSJ) dataset. We then input an audio utterance whose ground-truth transcription is given in the first row of Table 1. Four extracted beam candidates are listed in the following rows, from the best to the worst. Except beam 0, beam 1 to 3 all make some mistakes compared with the ground-truth. In this case, a language model is supposed to give a high score for beam 0 whereas low scores for beam 1 through 3. However, we observe that the scores of beam 2 and 3 are not sufficiently smaller than beam 0.

Table 1: Four beam candidates for an utterance in WSJ dataset. Mistakes are annotated in red.

| | sentence | score |
|---|---|---|
| true | user fees simply could not keep up with the soaring costs of loans and construction | $-81.42$ |
| 0 | user fees simply could not keep up with the soaring costs of loans and construction | $-81.42$ |
| 1 | user fee simply could not keep up with the soaring costs of loans and construction | $-84.87$ |
| 2 | usser fees simply could not keep up with the soaring costs of loans end construction | $-81.58$ |
| 3 | usser fees simply could not keep up with the soaring costs of loan end construction | $-80.34$ |

We denote the $i$-th ground-truth sentence as $\mathbf{x}_i$ where $i = 1, \ldots, N$; correspondingly, the $j$-th beam candidate as $\mathbf{x}_{i,j}$, where $j = 1, \ldots, B$ and $B$ is the beam size. Without loss of generality, we assume that these $B$ candidates all differ from the ground-truth by some mistakes/incompetences. The NLM is desired to assign big log-likelihoods for the $\mathbf{x}_i$'s but small ones for the $\mathbf{x}_{i,j}$'s.

## 3.1 A Straightforward Formulation

A straightforward way is to adopt the following objective:

$$\min_{\theta} \frac{1}{N} \sum_{i=1}^{N} \left( -\log p_\theta(\mathbf{x}_i) + \frac{1}{B} \sum_{j=1}^{B} \log p_\theta(\mathbf{x}_{i,j}) \right). \tag{2}$$

Similar formulation is also seen in Tachioka & Watanabe (2015), where they only utilize one beam candidate, *i.e.*, $B = 1$. The idea is to maximize the likelihood on the positive samples, and at the same time, minimize the likelihood on the negative samples. Optimization can be carried out by mini-batch stochastic gradient descent (SGD). Each iteration, SGD randomly samples a batch of $i$'s and $j$'s, computes stochastic gradient w.r.t. $\theta$, and takes an update step. However, a potential problem with this formulation is that the second term (corresponding to the negative samples) may dominate the optimization. Specifically, the training is almost always driven by the negative $x_{i,j}$'s, but does not effectively enhance the discrimination. We illustrate this fact in the following experiment.

Using the aforementioned ASR system, we extract 256 beam candidates for every training sample in the WSJ dataset. As a baseline for beam rescoring, a conventional NLM is trained on a large corpus, i.e. common-crawl [1]. From the pre-trained baseline NLM, we warm start the training and apply SGD to optimize the objective in Eq. (2), with a mini-batch size of 128. The training loss is shown in Figure 1a. We observe that the learning dynamic is very unstable. Using the trained model, we want the ground-truth sentences to have larger scores than the beam candidates. Therefore, we inspect $\log p_\theta(\mathbf{x}_i) - \log p_\theta(\mathbf{x}_{i,j})$, the *margin* between the scores of a ground-truth and a candidate. In Figure 2a, we histogram the margins for all the $i, j$'s in a dev set. The distribution appears to be symmetric around zero, which indicates poor discrimination ability. Given these facts, we conclude that the straightforward formulation in Eq. (2) is not effective.

---

[1] http://web-language-models.s3-website-us-east-1.amazonaws.com/wmt16/deduped/en-new.xz

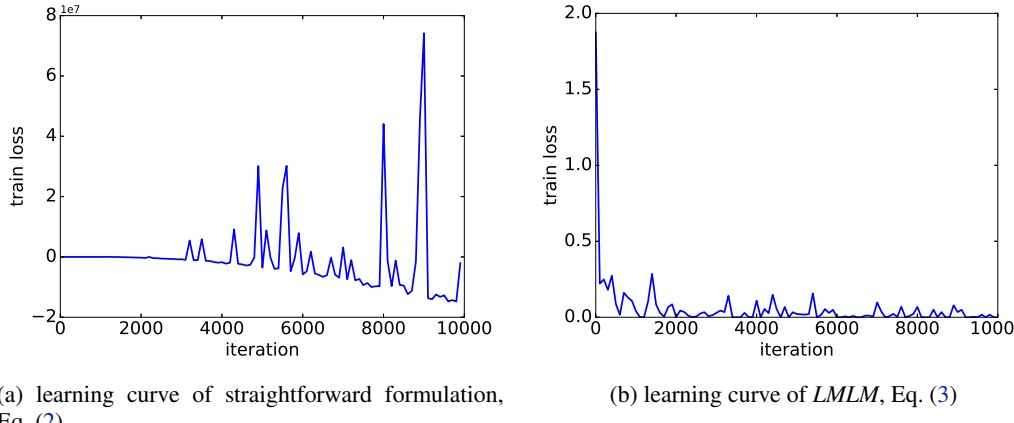

(a) learning curve of straightforward formulation, Eq. (2)

(b) learning curve of *LMLM*, Eq. (3)

Figure 1: Training losses of (a) straightforward and (b) large margin formulation

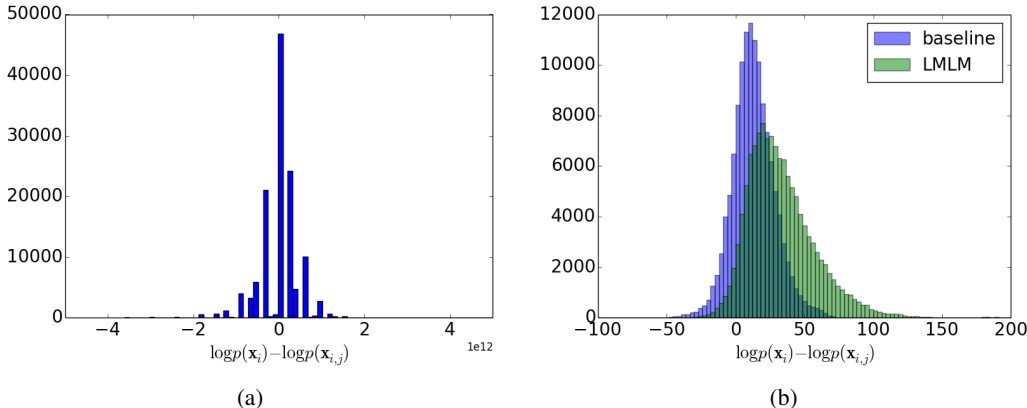

(a)

(b)

Figure 2: Histogram of $\log p(\mathbf{x}_i) - \log p(\mathbf{x}_{i,j})$, the margin between the log-likelihoods of ground-truth sentences and (erroneous) beam candidates. The more positive, the more the discrimination. (a) Straightforward formulation; (b) *LMLM* compared with baseline NLM (a conventional language model trained on common crawl)

## 3.2 LARGE MARGIN FORMULATION

To effectively utilize all the negative beam candidates, we propose the following objective,

$$\min_{\theta} \frac{1}{NB} \sum_{i=1}^{N} \sum_{j=1}^{B} \max \left\{ \tau - (\log p_{\theta}(\mathbf{x}_i) - \log_{\theta}(\mathbf{x}_{i,j})), \, 0 \right\}, \tag{3}$$

where $\log p_{\theta}(\mathbf{x}_i) - \log_{\theta}(\mathbf{x}_{i,j})$ is the margin between the scores of a ground-truth $\mathbf{x}_i$ and a negative candidate $\mathbf{x}_{i,j}$. The hinge loss on the margin encourages the log-likelihood of the ground-truth to be at least $\tau$ larger than that of the "bad" candidate. We call the above formulation Large Margin Language Model (*LMLM*).

We repeat the same experiment in section 3.1, but change the objective function to Eq. (3). We fix $\tau = 1$ across the paper. Figure 1b shows the training loss, which steadily decreases and approaches zero rapidly. Compared with the learning curve of naive formulation (figure 1a), the large margin based training is much more stable. In Figure 2b, we also examine the histogram of $\log p_{\theta}(\mathbf{x}_i) - \log p_{\theta}(\mathbf{x}_{i,j})$, where $p_{\theta}(\cdot)$ is the language model learned by *LMLM*. Compared with the histogram by the baseline NLM, *LMLM* significantly moves the distribution to the positive side, indicating more discrimination.

### 3.3 Enhanced Large Margin: Using Ranking Information

In most cases, all beam candidates are imperfect. It would be beneficial to exploit the information that some candidates are relatively better than the others. We consider ranking them according to some metrics w.r.t. the ground-truth sentences. For ASR, the metric is edit distance, and for SMT, the metric is BLEU score. We define $\mathbf{x}_{i,0} \triangleq \mathbf{x}_i$ and assume that the candidates $\{\mathbf{x}_{i,j}\}_{j=1}^{B}$ in the beam are sorted such that

$$\text{editDist}(\mathbf{x}_i, \mathbf{x}_{i,j-1}) < \text{editDist}(\mathbf{x}_i, \mathbf{x}_{i,j})$$

for ASR, and

$$\text{BLEU}(\mathbf{x}_i, \mathbf{x}_{i,j-1}) > \text{BLEU}(\mathbf{x}_i, \mathbf{x}_{i,j})$$

for SMT. In other words, $\mathbf{x}_{i,j-1}$ has better quality than $\mathbf{x}_{i,j}$.

We then enforce the "better" sentences to have a score at least $\tau$ larger than those "worse" ones. This leads to the following formulation,

$$\min_{\theta} \frac{1}{NB(B-1)/2} \sum_{i=1}^{N} \sum_{j=0}^{B-1} \sum_{k=j+1}^{B} \max\left\{\tau - (\log p_{\theta}(\mathbf{x}_{i,j}) - \log_{\theta}(\mathbf{x}_{i,k})),\ 0\right\}. \tag{4}$$

Compared with *LMLM* formulation Eq. (3), the above introduces more comparisons among the candidates, and hence more computational cost during training. We call this formulation *rank-LMLM*.

## 4 Experiments

In this section, we study *LMLM* and *rank-LMLM* through extensive experiments on ASR and SMT. We demonstrate that both *LMLM* and *rank-LMLM* significantly outperform a baseline language model (*baseline-LM*) and two other domain adapted models.

The *baseline-LM* architecture, shown in Figure 3, starts with a 2048 dimensional embedding layer, followed by two LSTM layers, each with 2048 nodes. The LSTM hidden states are then projected down to dimension 512. Finally, a softmax layer with 400K dimensional output is appended to produce a distribution over the vocabulary. The huge vocabulary size incurs a large computational cost, and we use sampled softmax technique (Jean et al., 2014) to accelerate training. This type of neural architecture has been shown effective in conventional language modeling tasks (Jozefowicz et al., 2016). We trained the *baseline-LM* on the common-crawl corpus. Common-crawl corpus has a vocabulary size about 400K. This large vocabulary ensures a small out of vocabulary (OOV) rate for the used ASR and SMT datasets, details of which are summarized in Table 2. The *baseline-LM* achieves a reasonably good perplexity of 110 on a dev set with 400K sentences, significantly outperforming a 5-gram model, which has a dev perplexity of about 300.

Table 2: Statistics of the used datasets

| dataset | common-crawl | WSJ | Fisher | IWSLT |
|---|---|---|---|---|
| task | *baseline-LM* | ASR | ASR | SMT |
| # train | 1.1G | 37.4K | 2.1M | 133.3K |
| # dev | 400K | 503 | 1,000 | 1,533 |
| # test | N/A | 333 | 4,458 | 1,268 |
| OOV | 0% | 0.28% | 0.05% | 1.07% |

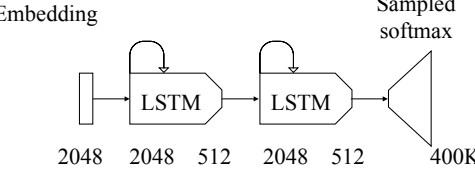

Figure 3: Architecture of *baseline-LM*

### 4.1 Experimental Protocol

The experimental setup is as follows. First we train an ASR/SMT model on the training set. Then we extract $B$ beam candidates for every training sample. This beam set, together with the corresponding ground-truth text, are used as the training data for *LMLM* and *rank-LMLM*. We then re-score the beams by linearly combining ASR/SMT and language model scores. The combination weight

is found by optimizing the WER/BLEU on the dev set. Finally, WER/BLEU on the test set are reported.

For comparison, we also include two other approaches that adapt the *baseline-LM* to the text in the specific task. One way is to fine-tune the baseline NLM using the task-specific text but still under the minimum-perplexity objective (Kim, 2014). We call it *refine-LM*. The other way is to train a smaller NLM from scratch on the task-specific data, and then linearly interpolate with the *baseline-LM* (Hsu, 2007). We call it *interp-LM*. In all the experiments with *LMLM* and *rank-LMLM*, we set $\tau = 1$. For *rank-LMLM*, since the total number of pairs is huge, we randomly sample $20\%$ of them.

### 4.2 ON THE IMPORTANCE OF WARM-STARTING

An advantage of *LMLM* (and *rank-LMLM*) is being able to utilize huge unsupervised data. This is achieved by warm-starting the *LMLM* with a conventional language model trained on any unsupervised corpus. However, one would doubt why it is necessary to warm-start, since *LMLM* might easily learn to make binary decisions on pairs of "good" and "bad" sentences. Interestingly, we show that warm-starting is very important. Without warm-starting, the *LMLM* training will be stuck to "bad" local minimal and cannot generalize well. Earlier works in various applications (Hinton & Salakhutdinov, 2006; Erhan et al., 2010) have observed similar behavior and suggested using unsupervised model to warm-start supervised training. The observation on *LMLM* is yet another supportive evidence, elaborated in the following experiment.

Using the same ASR system in Section 3.1, we extract 64 beam candidates for every training utterance in the WSJ dataset. We then train *LMLM* either with or without warm-starting from the *baseline-LM*.

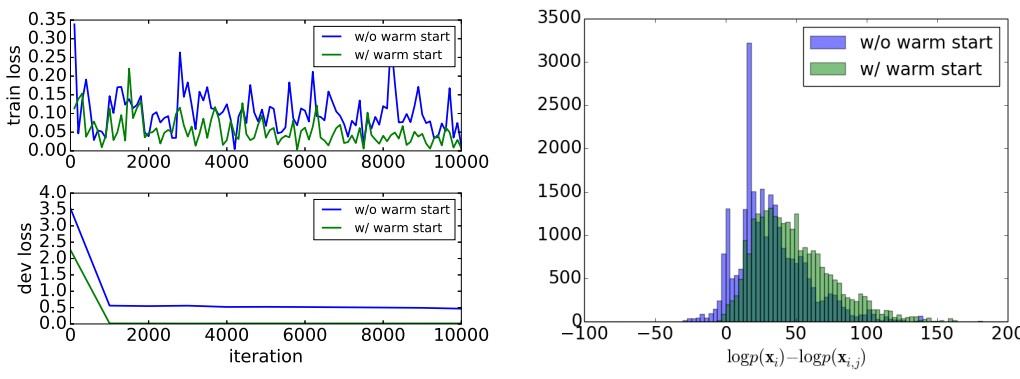

(a) training and dev loss with/without warm starting     (b) histogram of margin with/without warm starting

Figure 4: Compare the training of *LMLM* wither with or without warm starting

Figure 4a compares the learning curves for both cases. With warm starting, training and dev losses start to decay from smaller values, approaching zero at convergence. The stronger generalization ability is further illustrated in Figure 4b. Figure 4b shows the histogram of the margins between the scores for the ground-truths and beam candidates in the dev set. The more positive the margin is, the more the separation between positive and negative examples. In the warm-started case, the distribution is shifted rightwards, indicating more discrimination.

### 4.3 WSJ EXAMPLE REVISITED

We revisit the example in Table 1, in order to understand why *LMLM* and *rank-LMLM* can work. We estimate the language model scores using *LMLM* and *rank-LMLM*. Scores are listed in Table 3, in comparison with those by the *baseline-LM*, which we have seen in Table 1. Numbers in the brackets are the *margins*. A large positive margin indicates effective identification of the erroneous sentence. Overall, with *LMLM* and *rank-LMLM*, the margins become significantly more positive.

Table 3: Scores for the four beam candidates in table 1

| | sentence | scores (margin) | | | | | |
|---|---|---|---|---|---|---|---|
| | | baseline-LM | | LMLM | | rank-LMLM | |
| true | user fees . . . loans and . . . | −81.42 | | −86.48 | | −104.21 | |
| 0 | user fees . . . loans and . . . | −81.42 | (+0.00) | −86.48 | (+0.00) | −104.21 | (+0.00) |
| 1 | user fee . . . loans and . . . | −84.87 | (+3.45) | −89.82 | (+3.34) | −110.21 | (+6.00) |
| 2 | usser fees . . . loans end . . . | −81.58 | (+0.16) | −112.38 | (+25.90) | −124.54 | (+20.33) |
| 3 | usser fees . . . loan end . . . | −80.34 | (−1.08) | −111.47 | (+24.99) | −127.43 | (+23.22) |

Table 4: WER and CER on WSJ test set

| | no re-score | baseline-LM | refine-LM | interp-LM | LMLM | rank-LMLM |
|---|---|---|---|---|---|---|
| WER | 7.58 | 6.89 | 6.73 | 6.80 | 5.67 | **5.53** |
| CER | 2.72 | 2.67 | 2.58 | 2.65 | 2.24 | **2.19** |

Table 5: WER and CER on Fisher test set (SWBD and CallHome)

| | no re-score | baseline-LM | refine-LM | interp-LM | LMLM | rank-LMLM |
|---|---|---|---|---|---|---|
| WER | 28.38 | 27.57 | 26.89 | 27.58 | 26.01 | **25.99** |
| CER | 11.66 | 11.56 | 11.49 | 11.56 | 11.32 | **11.30** |

Table 6: BLEU scores on IWSLT test set

| | no re-score | baseline-LM | refine-LM | interp-LM | LMLM | rank-LMLM |
|---|---|---|---|---|---|---|
| BLEU | 14.18 | 14.18 | 14.18 | 14.18 | 15.51 | **15.80** |

More interestingly, *rank-LMLM* is able to assign larger score for beam 2 than beam 3, showing more selectivity than *LMLM*.

We also notice that all the scores by *LMLM* and *rank-LMLM* are smaller than those by the *baseline-LM*, since the proposed methods are not guided by the conventional max-likelihood objective. Compared with *LMLM*, *rank-LMLM* scores are even smaller. This is due to more pairwise constraints imposed in training, which makes *rank-LMLM* deviate even more from the max-likelihood objective. However, we argue that the max-likelihood training is not well aligned with beam re-scoring purpose, which we shall see in the reported WERs soon.

## 4.4 EXPERIMENTS ON ASR

In this section, we apply the proposed methods for ASR tasks, and report WERs and CERs on test sets. The datasets we used are WSJ and Fisher, whose statistics are summarized in the 3rd and 4th columns of Table 2. Note that for Fisher task, we follow the standard setup (Povey et al., 2016), which evaluates on the hub5 set, including two subsets (SWBD and CallHome). In total, there are 4,458 utterances for test.

The ASR models for WSJ has one convolutional layer, followed by 5 layers of RNNs. The ASR model for Fisher has two convolutional layers, followed by 6 layers of GRUs. The final hidden representations are input into fully connected layers and then the models are trained using CTC loss (Graves et al., 2006). Note that the ASR models are trained using only in-domain data. That is, the training utterances of WSJ or Fisher respectively. Using a language model during decoding may significantly improve the performance. Therefore, during decoding, we also applied an n-gram model learned from the in-domain training text. The beam search decoder has a beam width of 2000. The top-1 beam candidates give us strong baseline WERs, e.g., for WSJ task, the test WER is 7.58.

The extracted beams candidates are then re-scored using the *baseline-LM*, *refine-LM*, *interp-LM*, *LMLM* and *rank-LMLM*. Training of these language models follow the experimental protocol in section 4.1. We report the WERs of the rescored top-1 candidates on the test set. The same exper-

iment is repeated on WSJ and Fisher tasks. Results are listed in Table 4 and Table 5 respectively. Bold numbers are the best and italics are runner-ups. Among all the language models for re-scoring, *rank-LMLM* achieves the best WER and CER, and *LMLM* is also very competitive with *rank-LMLM*. They both significantly outperform the other methods, all of which are generative language models. Based on these observations, we argue that a language model has to be adapted in a way that is suitable for re-scoring purpose in supervised tasks, rather than just maximizing likelihood.

## 4.5 EXPERIMENTS ON SMT

*LMLM* and *rank-LMLM* are general in the sense that they can be applied to any re-scoring problem on text data. In this section, we show a proof-of-concept example on an SMT task. We experiment with IWSLT15 Vietnamese-to-English dataset[2]. We use "tst2012" as dev set and "tst2013" as test set. The dataset is simplified by decapitalizing all the words and removing punctuations. This results in a vocabulary size of about 44K, and only a tiny fraction of them are out of the vocabulary of common crawl. Details of the cleaned dataset are listed in the last column of Table 2.

We train an SMT model based on the attention mechanism in Luong et al. (2015). The encoder and decoder both have two layers of LSTMs, each with 128 activations. We follow the experimental protocol outlined in Section 4.1. Table 6 reports the BLEU scores on the test set. *LMLM* and *rank-LMLM* both significantly outperform the other methods. In contrast, all the generative language models (3rd to 5th column in Table 6) do not improve upon the case without re-scoring. In fact, for these three cases, we found that the best weight to combine the language model is zero, meaning that the language model is not providing any complementary information to the SMT model. A key point to understand this phenomenon is that the decoder in a seq-to-seq model implicitly works as a conventional language model (Sutskever et al., 2014). This results in very grammatical beam candidates, but their qualities differ in some semantic sense. A generative language model acts more or less like a grammar checker, and has no power of discrimination in the semantic field.

## 5 CONCLUSION

Conventional language models are guided by minimizing perplexity, and they are generative models. This work proposes an approach to enhance the discrimination ability of language models. It is trained end-to-end by maximizing the margin between "good" and "bad" (in a task-specific sense) sentences. The method is general and can be applied to various tasks that require re-scoring of text data. Experiments on ASR and SMT have shown a consistent gain over several baselines. These facts argue that min-perplexity is not necessarily an appropriate guideline when we want to apply language models in some supervised learning problems. A future direction is to apply the proposed method to conversation generation. The goal is to discriminate between boring (e.g., "I don't know") and informative replies, thus deprecating the former. Another interesting future work is to apply the *LMLM/rank-LMLM* to lattices during decoding.

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
