# OpenReview forum: "Large Margin Neural Language Models"
_ICLR.cc/2018/Conference — Reject_

### Official Review · AnonReviewer2 · 2017-11-20
**Interesting methods but weak baselines (and difficulty of replication) dilute impact of experimental results**

**Rating:** 5
**Confidence:** 4

**Review:**

This paper presents two methods for imposing a margin on discriminative loss functions, one which uses the margin between the reference transcription and alternatively hypothesized transcriptions (LMLM), and another which compares all alternative candidates and uses a margin between those with a better system objective (WER or bleu) and those with a worse system objective (rank-LMLM).  Some interesting results on the development set show the importance of things like warm starting on large language model training data.  The methods presented here could be of interest to those training language models for use in specific systems, and the paper reads reasonably clearly.

The principal shortcoming of the paper is that there was essentially no effort to establish that the baseline systems that are being improved through reranking via these methods are decent baselines for such a use, or to really specify these systems in a way that would allow for replication of the results being presented in the paper.  Sufficient specification of the exact training data and procedure is standard in papers that purport to establish methods to improve upon such baselines, yet such information is sorely lacking in this paper.  Further, the speech data sets, Fisher and Wall St. Journal, have what would seem to be very high word error rates versus what should be possible with standard open-source speech recognizers such as Kaldi.  For example, by referencing a page that attempts to establish the state of the art on standard data sets (https://github.com/syhw/wer_are_we), we can find links to papers by Povey et al (http://www.danielpovey.com/files/2016_interspeech_mmi.pdf) and the Deep 2 paper in your citations, which themselves include baselines from other papers that cut the error rate in half versus even your best scoring systems, let alone your baselines.  Similarly, your Bleu score on Vietnamese to English translation is way below what were reported (even by the organizer baseline) for the IWSLP conference where the data became available: https://github.com/magizbox/underthesea/wiki/SOTA-Machine-Translation:-IWSLT-2015

Granted, the competing systems also were outperformed by the organizer baseline for that task at IWSLT 2015, but not by the degree to which your system is.  Again, your best performing system (using your new methods) has performance far below the worst reported competing system.  The cavalier presentation of specific details regarding your baseline systems (which is critical for any sort of replicability) and the uniformly weak performance of these systems relative to widely reported results, leads me to discount the probability that your methods would actually result in improvements on truly solid baselines.  I would have preferred one domain experiment carried out with appropriately rock solid documentation of the ball-park competitive baseline system to these results.

Overall, the method is interesting and the dev set experiments were informative, but ultimately the experiments were not.

Revision:  Having read the author response for this paper, I am encouraged by the updated baseline for WSJ and the additional explication about the competing Fisher systems.  The additional information about the systems included in section 4.4 is pretty nominal, though, and I worry about the ability of others to replicate these results.  I would have felt better about the results if there were reported results from other papers included here, instead of the authors' attempt to create a baseline from the given data, which may or may not (as we have seen) represent a strong enough baseline from which to draw conclusions.  That is to say that I still have some reservations (though less than I had before).  I am including this additional information in my review, for use by the area chair, but otherwise leaving my assessment as-is.

---

> ### Author Response · Authors · 2018-01-05
> **Experiments with strong baseline and sufficient gain is observed**
>
> “Overall, the method is interesting and the dev set experiments were informative” -- We are glad that the reviewer likes our proposed method.
>
> “no effort to establish that the baseline systems …”, “the speech data sets …, have what would seem to be very high word error rates…”, “… leads me to discount the probability that your methods would actually result in improvements on truly solid baselines” -- The major concern is whether the gain of re-ranking will still exist if a stronger baseline ASR/SMT system is used. To address this, we have carefully re-conduct the experiments on WSJ dataset, and obtained a stronger baseline of 7.58 WER. Based on that, LMLM and rank-LMLM further reduces the WER to 5.67 and 5.53 respectively. This improvement is bigger than all the other comparative methods. For the Fisher task, people often replace some special words (with those from a template) before calculating WER. This will drastically reduce WER. Here since we want to focus on the method itself, we did not apply this trick. That is why the WER is higher than the reported ones in the other literature. Details regarding the setup of the baseline and new results can be found in our revised draft (2nd paragraph of section 4.4).
>
> We are grateful that the reviewer points us to a very useful webpage of state-of-art WERs. However, it should be noticed that some of the results there are obtained from ASR systems trained with more out-of-domain data, e.g., the 3.60 WER on WSJ in Deep Speech 2, uses 11.9K hours of training utterances. Whereas in our work, we restrict all the ASR systems trained to use only the in-domain training sets. This will surely degrade the WER. For example, https://arxiv.org/pdf/1707.07413.pdf reports a WER of 10.08 (Tab. 5) on WSJ test set with an in-domain setup. Since our new baseline already beats this reported in-domain result, we consider the gain after applying LMLM (and rank-LMLM) as convincing. The reviewer also points out the lack of effort to “specify these systems in a way that would allow for replication of the results”. Indeed, more details should be provided, e.g., the in-domain setup should be emphasized. We have detailed the data and training procedure in our revised draft (2nd paragraph of section 4.4), which should enable reproducibility.
>
> “… Bleu score on Vietnamese to English translation is way below what were reported…” -- We are not able to finish the experiment on machine translation (MT) before the deadline of rebuttal. We will include the stronger results in final version.

---

### Official Review · AnonReviewer3 · 2017-11-30
**Careful application of large margin in neural language model, good empirical results on ASR and MT tasks**

**Rating:** 7
**Confidence:** 5

**Review:**

The main contribution of this paper are:
(a) replacing the typical maximum likelihood criterion in neural language model training with a discriminative criterion,
(b) propose two large margin criterion -- difference in likelihood and difference in rank (WER or BLUE ordered) hypotheses,
(c) demonstrate performance gains two standard tasks -- an ASR task on Wall Street Journal (small task) and an MT task.

In addition, they provide examples in Figure (1) and (2) that illustrate the effect of the cost function on training. Their illustration in Figure 4 is also helpful in seeing the impact of using a warm start with a generative model.

---

> ### Author Response · Authors · 2018-01-05
> **Glad that the reviewer likes our paper**
>
> We are glad that the reviewer appreciates our surgery/case-study of the proposed method. Indeed, one of our contributions is to not only show the improvements in various tasks, but also look into why the method would work. We believe this will help other researchers better understand end-to-end “deep” re-ranking of text.

---

### Official Review · AnonReviewer1 · 2017-12-05
**This paper proposes to use a discriminative objective in training NN language models with application in a re-scoring framework for ASR and MT tasks.**

**Rating:** 5
**Confidence:** 5

**Review:**

A large margin , end to end language model that uses a discriminative objective function is proposed. The proposed objective imposes a hinge loss on the margin to ensure that the ground truth is at least  some fixed amount larger than the imposter. A variant on this, which also incorporates the ranks of the imposters sorted by a metric such as edit distance or BLEU metric with respect to the ground truth is also introduced.

The paper is missing some of the original references to a discriminative LM (DLM) as well as  references to the use of a NN LM directly in decoding (presented in ICASSP and Interspeech conferences over the last 5 years). For example, H.-K. J. Kuo, E. Fosler-Lussier, H. Jiang, and C.-H. Lee, “Discriminative training of language models for speech recognition,” in Proc. ICASSP,
vol. 1, 2002, pp. 325–328.

Have you considered the widely-used NCE method to handle the large vocabulary?

The dev perplexity quoted in Section 4 for a 5 gram LM is very high.  Also Table 4 and Table 5 on WSJ and  FIsher  show baseline experiments that are quite far away from the state-of-the-art in these tasks. Even if you assume that you use the simplest possible acoustic model and/or an open source tool kit for the decoder,  these error rates are high (WSJ error rates are lower than 10%, not 16.7%). Even if the LM is trained on the common-crawl corpus, it has  a very low OOV rate, and fine tuning on the tasks only lowers it b t 1%.  For reference,  please see papers from Saon et al., Seide et al, Povey et al, Yajie Miao et al in various ICASSP, Interspeech and arXiv papers. Comparisons with weak baselines can significantly color the conclusions. On the Fisher test set, the interpolated LM offers very little over the baseline LM in Table 5. This is contrary to what is observed in the literature.  There is not much difference between rankLM and LMLM as well to draw a clear conclusion between the two. Given that this is n-best rescoring, how are the N-best lists generated?  You state that they are extracted from  64 beam candidates, are they unique N-best lists?  Can this method be applied to lattices? What is the perplexity of all the language models corresponding to  Tables 4 and 5? This would have been useful to study in itself.

In the SMT tasks, the baselines reported seem to be far away from results presented in the literature on the IWSLT task (see http://workshop2015.iwslt.org/downloads/IWSLT_2015_EP_3.pdf)

While the proposed objective is interesting and meaningful for several conversational applications, as well as sentence modeling, the presented experimental results are not convincing.

---

> ### Author Response · Authors · 2018-01-05
> **Addressing most of the comments**
>
> We thank the reviewer for the constructive comments.
>
> “While the proposed objective is interesting and meaningful for several conversational applications …” -- We are glad that the reviewer appreciates our idea.
>
> “… missing some of the original references  …” – We have cited and discussed the discriminative training work in revised draft. See the 2nd paragraph in Related Work section.
>
> “Have you considered the widely-used NCE method to handle the large vocabulary?” – Indeed we have also considered NCE, but sampled softmax has also been shown to be a very competitive method, e.g., in https://arxiv.org/abs/1602.02410 (See also the code by those authors https://github.com/rafaljozefowicz/lm/blob/master/language_model.py#L99).
>
> “dev perplexity quoted in Section 4 for a 5 gram LM is very high” – This is indeed what it looks like when we train a 5-gram model using the kenlm package. This result is reproducible.
>
> “on WSJ and Fisher show baseline experiments that are quite far away from the state-of-the-art”. We have carefully re-conducted the experiments on WSJ dataset, and obtained a stronger baseline of 7.58 WER (before applying any language model to re-score). Based on that, LMLM and rank-LMLM further reduces the WER to 5.67 and 5.53 respectively. In the revised draft, we have included the details of experimental setup in the 2nd paragraph of section 4.4. Regarding the experiments on the Fisher task, people often replace certain special words before calculating WER. This will drastically reduce the WER. Here since we want to focus on the method itself, we did not apply this trick. That is why the WER is higher than the other reported ones.
>
> “There is not much difference between rankLM and LMLM …” – Indeed the difference is small, we suspect it is due to a too small test set. Experiments on even larger data set is for future work.
>
> “How are the N-best lists generated” – We generate the N-best list using beam search decoder. In our updated ASR experiments, an n-gram model is trained using in-domain training text. This n-gram model is utilized during decoding. Details can be found in the 2nd paragraph of section 4.4.
>
> “Are they unique N-best list” – For CTC models, it’s possible that some beam candidates duplicate each other. However, we removed the duplicates when we generate the pairs to train LMLM and rank-LMLM.
>
> “Can this method be applied to lattices” – Yes, we defer this to future work (Section 5).
>
> “What is the perplexity of all the language models corresponding to Tables 4 and 5” -- The perplexity of LMLM/rank-LMLM is bigger than a conventional language model, since they are not designed to maximize likelihood. We have observed and discussed this fact in the last paragraph of section 4.3.
>
> “In the SMT tasks, the baselines reported seem to be far away …” – We are trying to improve the SMT baseline (like we did for ASR), but not able to finish it before the rebuttal deadline. At this stage, the SMT experiment is more or less a proof of concept. We will include improved results in final version.

---

### Decision · Program_Chairs · 2018-01-29
**ICLR 2018 Conference Acceptance Decision**

**Decision:**

Reject

**Comment:**

Pros
-- Proposes an interesting margin-based training for LMs.

Cons
-- Experimental evidence is lacking, and it’s not clear if the baselines are strong enough.

Given the reviewer comments, especially regarding experimental evaluations, the AC recommends that the paper be rejected.